# The Complex Role of Chaperone-Mediated Autophagy in Cancer Diseases

**DOI:** 10.3390/biomedicines11072050

**Published:** 2023-07-20

**Authors:** Jing Liu, Lijuan Wang, Hua He, Yueying Liu, Yiqun Jiang, Jinfeng Yang

**Affiliations:** 1Department of Basic Medicine, School of Medicine, Hunan Normal University, Changsha 410013, China; liujing@hunnu.edu.cn (J.L.); hehua123@hunnu.edu.cn (H.H.); liuyy@hunnu.edu.cn (Y.L.); 2Department of Anesthesiology, Hunan Cancer Hospital/The Affiliated Cancer Hospital of Xiangya School of Medicine, Central South University, Changsha 410013, China; 15093142336@163.com; 3The Key Laboratory of Model Animal and Stem Cell Biology in Hunan Province, Hunan Normal University, Changsha 410013, China

**Keywords:** autophagy, chaperone-mediated, lysosome, cancer protein, degradation

## Abstract

Chaperone-mediated autophagy (CMA) is a process that rapidly degrades proteins labeled with KFERQ-like motifs within cells via lysosomes to terminate their cellular functioning. Meanwhile, CMA plays an essential role in various biological processes correlated with cell proliferation and apoptosis. Previous studies have shown that CMA was initially found to be procancer in cancer cells, while some theories suggest that it may have an inhibitory effect on the progression of cancer in untransformed cells. Therefore, the complex relationship between CMA and cancer has aroused great interest in the application of CMA activity regulation in cancer therapy. Here, we describe the basic information related to CMA and introduce the physiological functions of CMA, the dual role of CMA in different cancer contexts, and its related research progress. Further study on the mechanism of CMA in tumor development may provide novel insights for tumor therapy targeting CMA. This review aims to summarize and discuss the complex mechanisms of CMA in cancer and related potential strategies for cancer therapy.

## 1. Introduction

The synthesis and degradation of proteins determine cell growth and differentiation, playing important roles in the growth and development of organisms. The two pathways work in a coordinated manner to maintain cellular homeostasis [1]. As early as the 1970s, study indicated that there are two pathways for protein degradation: the ubiquitin–proteasome system pathway and the lysosome-mediated autophagy pathway [2].

Autophagy, which occurs in lysosomes containing multiple hydrolases, is a major mechanism for protein degradation in the intracellular lysosomal pathway, and its function is to degrade unnecessary components both inside and outside the cell. Autophagy contributes to the maintenance of cellular homeostasis, cellular quality control, the ability to defend against damage from inside and outside the cell, and the maintenance of cellular energy homeostasis, which plays a positive role in cell survival [3]. There are three types of autophagy: macroautophagy, microautophagy, and chaperone-mediated autophagy. Macroautophagy is the most common form of autophagy in mammals, and it occurs to maintain cytoplasmic health. In addition to hypoxia, inflammation, etc., it is more active under nutrient-starvation conditions [1]. In this type of autophagy, substrate proteins and organelles are sequestered within a pre-existing autophagosome. The autophagosome then fuses with the lysosome to form an autolysosome before protein degradation occurs. On the contrary, microautophagy does not involve autophagosomes, where cytosolic components are directly engulfed invaginations or protrusions of the lysosomal membrane, forming small vesicles that bud off protein clearance. Chaperone-mediated autophagy uses a chaperone-dependent selection mechanism to deliver components requiring clearance in the cell to the lysosome without forming vesicles [4]. The key to distinguishing the CMA pathway from the other two pathways is its selective and multi-step process. This review will focus on CMA.

The CMA degradation pathway starts with the selective recognition of substrate proteins containing pentapeptide motifs (KFERQ) by heat shock protein family A member 8 (HSPA8), which subsequently binds to lysosome-associated membrane protein type 2a (LAMP-2A) molecules on the lysosomal membrane and initiates multimerization to rapidly degrade substrate proteins after they enter the lysosome [5]. Within the intensive study of the process and mechanism of this action, there is increasing interest in the regulatory role of CMA in diseases, especially cancer. Early studies resulted in the original finding of the connection between CMA and cancer, that CMA is required for tumor growth. Kon M et al. demonstrated that inhibiting CMA effectively slows down xenograft growth, reduces the number of metastatic cancer cells, and induces a reduction in size in existing human lung cancer xenografts in *mice* [6]. Basal CMA activity is continuously enhanced in multiple cancer cell lines and human tumor biopsies, suggesting a link between CMA and the pathogenesis of multiple cancers [7,8].

In this paper, we will first outline the main features of the CMA pathway and its physiological functions, with a focus on its dual role in various types of cancer. We will then elaborate on the mechanisms of CMA action in cancer and finally discuss relevant strategies for cancer therapy. These findings may provide meaningful references for targeted therapy of cancer by studying the role of CMA.

## 2. Chaperone-Mediated Autophagy

### 2.1. CMA Components

#### 2.1.1. Substrate Proteins

The substrate protein is recognized and binds specifically to the CMA receptor. Early studies identified RNase A as the first CMA substrate; it was confirmed to contain the pentapeptide motif of KFERQ. These motifs were recognized by the chaperone molecule HSPA8, which targets them for the lysosome. Subsequently, researchers identified KFERQ motifs similar to the pentapeptide motif in other proteins. Therefore, identifying proteins containing KFERQ-like motifs is a natural way to mark CMA substrates. The consensus sequence usually contains one or two positively charged amino acids (Lys, Arg), one or two hydrophobic residues (Phe, Leu, Ile, or Val), and a negatively charged Glu or Asp, flanked by glutamine residues [9]. About 30% of cytoplasmic proteins are known to contain such a KFERQ-like sequence, and some substrate proteins can be modified post-translation to produce similar functional targeting motifs that become CMA substrates [10]. Changes in CMA substrate level can indicate altered CMA activity. As more substrates have been identified, the CMA pathway has become increasingly closely related to many cellular processes [4].

#### 2.1.2. Main Companion

HSPA8, which is constitutively expressed and distributed throughout the cytoplasm and lysosomes, plays a primary role in the CMA pathway as a core component required for targeting substrates for lysosomes [11]. HSPA8, a member of the Hsp70 family of heat shock proteins, has a substrate-binding domain and a nucleotide-binding structural domain. It is the only chaperone molecule identified to recognize and bind substrates containing KFERQ-like motifs. HSPA8 has multiple functions, such as binding to hydrophobic regions of the protein and assisting in protein folding [4]. In addition, other chaperones (Hsp40 and Hsp90) are involved in this pathway. The latter can stabilize the translocation complex formed after the substrate binds to the receptor protein, which needs to be unfolded with the help of the chaperone molecule and then translocated into the lysosomal membrane [12].

#### 2.1.3. CMA Receptors

LAMP-2A is a receptor protein on the lysosomal membrane for the CMA pathway and is one of the splicing variants of the *LAMP2 gene*. The other two variants, LAMP-2B and LAMP-2C share the lumenal structural domain that is common among the three, but they differ from the LAMP-2A type in the transmembrane and cytoplasmic tail regions [13]. LAMP-2A monomers on lysosomal membranes bind to substrate proteins and oligomerize to form translocation complexes, which revert to a monomeric state after substrate translocation. The stability of LAMP-2A is tightly regulated by lysosome-associated proteins, such as glial fibrillary acidic protein (GFAP)/eukaryotic elongation factor 1 alpha (*EF1 α*). GFAP interacts with *EF1 α*, whereas guanosine triphosphate (GTP) induces GFAP to release EF1 α, inhibits CMA, and promotes the disassembly of the multimeric LAMP-2A [14]. The expression level of LAMP-2A on the lysosomal membrane is directly correlated with the activity of the protein degradation pathway. The changes occurring in the lysosomal membrane with age lead to reduced levels of LAMP-2A, thus decreasing the activity of CMA in aging organisms [15]. Therefore, the stability of LAMP-2A is closely related to the level of CMA activity (Figure 1).

### 2.2. CMA-Related Physiological Functions

CMA plays a vital role in physiological processes, including protein balance, cell energy, the cell cycle, cell survival, immune response, and cell metabolism [16].

Under conditions of nutrient deficiency in organs and cells, the CMA pathway is activated in the body. However, under starvation conditions, the human body first activates the macroautophagy pathway. When macroautophagy reaches its maximum activity, CMA is activated [13]. When hungry, cells break down unnecessary proteins into free amino acids. Subsequently, amino acids can enter the amino acid cycle to generate energy and synthesize major proteins required for cell functioning to ensure basic life activities. For example, the carbon skeleton of gluconeogenesis comes from alanine and L-glutamine decomposed by proteins. Evidence has shown that upregulated CMA in rodents can be maintained at high expression levels for up to 3 days [3]. Because CMA has a unique selective mechanism, it regulates cellular energy by selectively degrading critical enzymes involved in lipid and glucose metabolism, thus reducing energy deficit. CMA participates in regulating glucose and lipid metabolism, and its targets for selective degradation usually include lipogenic enzymes and lipid droplet shell proteins in addition to glycolytic enzymes [17]. This suggests that CMA plays a major role in controlling metabolic pathways and cellular energy.

CMA has antioxidant effects. The latest research shows that under oxidative conditions, the activated CMA pathway is directly involved in the degradation of the oxidized protein Kelch-like ECH-associated protein 1 (Keap1) [18]. The long-term accumulation of damaged proteins can have a detrimental effect on the human body. CMA transports the protein to be recovered or the old organelle from the cytoplasm to the lysosome for degradation under corresponding physiological conditions. CMA controls intracellular quality by promptly cleaning up harmful proteins [5]. The control of cellular quality by CMA may also be achieved by eliminating excess subunits and helping to assemble protein complexes, in addition to performing protein degradation. One study has already demonstrated that CMA could target subunits in the catalytic core of the proteasome, directing them to the lysosome for degradation [13]. Regardless, it facilitates the basic survival of the cell.

Checkpoint kinase 1 (CHK1) is one of the substrates of CMA. CMA plays an important role in regulating the cell cycle in response to DNA damage by timely degrading target proteins associated with the regulation of the cell cycle [19]. For example, poly (ADP-ribose) polymerase 1 (PARP1) is an enzyme involved in DNA damage repair; Zhang et al. first revealed that CMA targets PARP1 to inhibit cell apoptosis through lysosomal degradation [20]. Similarly, CMA selectively degrades the negative regulatory factor of T-cell response signals and is involved in the regulation of immune responses by activating CD4 T-cells [21]. Moreover, CMA is involved in class II antigen presentation in professional antigen-presenting cells. Reduced expression of HSPA8 and Hsp90 in professional antigen-presenting cells attenuates the CMA pathway and antigen presentation [22]. This further indicates that CMA also plays a role in the biological immune response, but the mechanism of this process remains to be studied. With age, the immune response in the organism may be attenuated by a decrease in CMA activity; thus, improving the activity of CMA in T-cells and enhancing the immune function of senescent cells may reduce the incidence of cancer in the elderly population.

### 2.3. Regulation of CMA

First of all, due to the specificity of CMA, there are many regulatory mechanisms, and studying CMA activity regulation is conducive to finding focuses for disease treatment. The most studied and critical restriction point is the number and assembly of the substrate protein LAMP-2A on the lysosomal membrane [23]. The chaperone HSPA8 is involved in substrate targeting and translocation and is distributed both inside and outside the lysosomal membrane. The specific regulation activity of HSPA8 in CMA remains unknown, but the CMA pathway is incomplete without it.

CMA activity can be regulated by several signaling pathways, including the calcium-regulated phosphatase–NFAT pathway, which is essential for CMA activation in T-cells. This pathway also contributes to the CMA response to oxidative stress and was the first signaling mechanism identified in CMA activation. During T-cell activation, the generation of reactive oxygen species (ROS) promotes the nuclear translocation of NFAT1 and upregulates LAMP-2A expression, thus further promoting CMA activity [21]. Endoplasmic reticulum (ER) stress may affect CMA activation. Li et al. found that ER stress initially activates p38 mitogen-activated protein kinase (p38MAPK) in the lysosome, leading to a series of activities that activate CMA. They also demonstrated that various stressors triggering ER stress induce PKR-like ER protein kinase (PERK)-dependent activation and the recruitment of mitogen-activated protein kinase kinase 4 (MKK4) [24,25]. However, the mechanism of how ER stress sends signals to lysosomes to trigger the CMA pathway is still unknown.

Bandyopadhyay et al. identified two regulatory proteins, glial fibrillary acidic protein (GFAP) and eukaryotic elongation factor 1 alpha (*eEF1α*), which mediate the negative regulation of CMA via guanosine triphosphate (GTP) [26]. The presence or absence of GTP determines the binding of GFAP to the lysosomal membrane, which affects CMA activity to different degrees. In the absence of GFAP, the inhibitory effect of GTP on substrate binding is significantly weakened. The study demonstrated that signaling of retinoic acid receptor alpha (RARα) reduces CMA activity by inhibiting the transcription of LAMP-2A, and the finding was confirmed by the observation of increased protein degradation in lysosomes upon blocking of RARα [27]. RARα inhibitors may restore CMA activity in aging cells, and developing RARα inhibitors could provide a new approach to maintaining CMA activity.

Chava et al. detected increased expression of LAMP-2A in hepatocellular carcinoma (HCC) while identifying a compensatory mechanism between macroautophagy and CMA: the activation of CMA can replace the impaired macroautophagy pathway, leading to increased survival in cirrhotic hepatocellular carcinoma [28]. Interestingly, recent studies have shown increased activity of macroautophagy in senescent cells, suggesting that macroautophagy can compensate for the impaired CMA pathway [29]. In this compensatory mechanism, the ability to reduce the sensitivity of cells with low CMA activity to stressors and thus allow them to better exert autophagic function has not yet been explained. This crosstalk pattern between macroautophagy and CMA needs further investigation.

#### CMA and Tumor Microenvironment

The tumor microenvironment refers to the cell environment where tumor cells are located, including various stromal cells (fibroblasts, lymphocytes, and macrophages), immune cells, and extracellular components (growth factors, hormones, etc.) [30]. The tumor microenvironment alters the metabolism of tumor cells through metabolic stress, such as hypoxia and high levels of ROS, which can activate CMA pathways to regulate tumor cell development [31].

Research on the role of CMA in the tumor microenvironment shows that the CMA pathway in the tumor microenvironment can promote tumor growth. Glioblastoma cells interact with pericytes to increase the activity of CMA in pericytes through high levels of ROS produced by pericytes, thereby promoting the proliferation of tumor cells and eliminating the immune anticancer response [32]. The tumor-associated macrophage (TAM) is the predominant type of immune cell in the tumor microenvironment. Wang et al. found that tumor cells act on TAMs, upregulating the expression of LAMP-2A, which eliminates antitumor immunity [30]. M2 tumor-associated macrophages in the tumor microenvironment can secrete the proinflammatory cytokine IL-17 (interleukin-17), which induces drug resistance in tumor cells by activating the CMA pathway and thus reducing apoptosis of HCC cells [33]. Hypoxia-inducible factor-1 subunit alpha (HIF-1α), which is overexpressed in the tumor microenvironment and is a substrate of CMA, is involved in cell cycle blockading and enables tumor cells to adapt to hypoxic environments [34]. Early studies revealed that the natural product manassantin A (ManA) is an inhibitor of HIF-1α [35]. New research suggests that ManA can degrade HIF-1α via the autophagy–lysosome pathway, but the specific mechanism by which it may participate in the CMA pathway is unclear. In addition, ManA was shown to activate CMA by inhibiting Hsp90, thus inhibiting the growth of cancer cells under hypoxic conditions [36]. Therefore, targeting the CMA pathway-related factors in the tumor microenvironment can provide feasible strategies for cancer treatment.

## 3. The Dual Role of CMA in Cancer

Recent studies have together suggested a dual role of CMA in cancer. In short, the anticancer functioning of CMA in healthy cells is changed in cancer cells [37]. The reason why CMA can degrade the key proteins required for tumorigenesis in normal tissues to prevent malignant transformation of cells but also protect the integrity of cancer cells in malignant transformation tissues and provide energy for the growth of cancer cells is not yet clear. The mechanism of this dual effect may be related to the different stages of tumor formation, which warrants further investigation.

### 3.1. Anticancer Effect of CMA

Although the decline in CMA activity and its functional impediment was first revealed in aging, later studies have gradually indicated an intricate link between CMA and cancer pathogenesis because of the close association between aging and cancer [38]. The anticancer effects of CMA on healthy tissues are mainly manifested in protein balance, metabolic regulation, stress response, genomic stability, degradation of oncogenes, and immunogenic adaptation. Among them, the anticancer ability of CMA is at least related to its selective inhibition of the functioning of tumor-related proteins (such as the proto-oncogene protein MDM2) [39,40].

The CMA pathway regulates cancer cell growth by remodeling the proteome: downregulating CMA activity prevents the degradation of proteins required for cancer development, whereas upregulating its activity selectively degrades proteins that inhibit cancer cell growth. Analysis of the proteomics and transcriptomics of glioblastoma revealed that CMA engages in various processes affecting glioblastoma stem cell (GSC) activity. It clarified that low activity of LAMP-2A decreases the mitochondrial activity of GSC, which leads to reduced ATP production, thus leaving cancer cells lacking the energy required for survival [41]. In normal cells, CMA triggers a selective mechanism that blocks the action of oncogenic proteins and prevents cells from progressing toward tumors. From this, it is thought that CMA plays a role in protein quality control for cancer inhibition.

Existing studies have shown that this triggering mechanism of CMA can also exert anticancer effects by degrading the corresponding promoter of tumor occurrence [42]. As mentioned earlier, the downregulation of the lysosomal oncogene protein MDM2 proposed by Lu, T.L., and others is the first example of the CMA pathway inhibiting tumors [40]. Similarly, in fibroblasts, Gomes et al. (2017) showed that CMA regulates intracellular levels of the proto-oncogene *MYC* by promoting its ubiquitination and degradation in a degradative manner involving protein phosphatase 2A (CIP2A), thereby eliminating its oncogenic activity [43]. Blocking the CMA pathway leads to increased and relatively stable levels of *MYC*, exacerbating its phosphorylation in cells with impaired CMA functioning, resulting in significant changes in tumorigenesis metabolic capacity. Therefore, reducing and preventing the accumulation of *MYC* is crucial in preventing malignant transformation in healthy cells.

Many other tumor promoters (such as the epidermal growth factor receptor pathway substrate 8 (Eps8), hexokinase 2 (HK2), galactose lectin-3, etc.) are degraded via the CMA pathway to deactivate them and inhibit tumor growth [44]. In 2010, Welsch T et al. demonstrated that CMA is involved in the degradation of the substrate protein Eps8 in the oncogenic protein epidermal growth factor receptor pathway; although the mechanism of action has not been elucidated in cancers (squamous cell carcinoma, pancreatic cancer), it still provides a molecular mechanism for the treatment of solid malignancies [45,46]. In addition, the CMA pathway exerts anticarcinogenic effects by influencing the glucose metabolism pathway. An enzyme in glucose metabolism, HK2, is necessary for the growth and maintenance of many tumors due to tumor cells being active in metabolism [47]. HK2 is a substrate of the CMA pathway. Degradation of HK2 via the CMA pathway can impede the process of glucose metabolism, thereby compromising the metabolism of cancer cells. A study on ovarian cancer found that degradation of HK2 occurred following the recognition of HK2-exposed KFERQ motifs by HSPA8 under conditions of cellular glucose deficiency [48]. *Tp53* encodes the *p53* tumor suppressor, yet mutations in *Tp53* contribute to the growth of cancer cells. Activated CMA targets mutant *p53* for lysosomal degradation to inhibit cancer progression. CMA-mediated degradation of mutant *p53* in cancer cells is a new way to treat *Tp53*-type cancer [49]. In addition, CMA plays a role in genomic stability and can protect healthy cells from malignant transformation during effective DNA repair processes. The role of CMA in genome quality control is achieved through the degradation of Chk1. Chk1 is a CMA substrate that regulates DNA damage-induced cell cycle arrest. Therefore, considering that inhibition of Chk1 during DNA damage can hinder the self-repair of tumor cells, the goal of eliminating tumor cells can be achieved [19,50]. Combined with the above, it is conceivable that exploring the activity of CMA in preventing cancer occurrence could have potential and far-reaching significance for addressing the much-needed cancer prevention and treatment problems facing humanity.

When cancer cells form new tumors, they leave the primary tumor and metastasize to other organs or tissues. CMA is thought to be possibly linked to tumor metastasis, as it has been shown that blocking LAMP-2A in the CMA pathway can significantly reduce the tumor metastatic ability of lung cancer cells [6]. As Xuan et al. showed in colorectal cancer, inhibiting CMA activity can decrease tumor metastasis and promote drug sensitivity [51]. In addition, CMA is also involved in the degradation of 17β-hydroxysteroid dehydrogenase type 4 (HSD17B4), a multifunctional protein that is significantly overexpressed in breast cancer, thereby attenuating cell migration [52]. Blocking the role of CMA in tumor metastasis can prevent cancer from forming new lesions in the body, and we believe this is an emerging hot topic for targeted cancer treatment in the future.

### 3.2. Cancer-Promoting Effect of CMA

Although CMA can timely prevent the accumulation of intracellular and extracellular damage in healthy cells, its protumor role in malignantly transformed cells is dominant. Several studies have shown that CMA activity may be upregulated after tumor formation, mainly through energy regulation, cell cycle regulation, and oxidative stress, increasing tumor survival and contributing to tumor metastasis [11]. The constitutive activity of CMA was shown to be expressed in most tumor cells after malignant transformation. LAMP-2A overexpression has been found in cancer cells, such as hepatocellular carcinoma, colorectal cancer, and breast cancer, which grow favorably by increasing the substrate levels of the CMA pathway [23,53,54]. Recently, Sohn et al. found that LAMP-2A and HSPA8 were highly expressed in ovarian cancer stem cells, indicating that CMA may regulate ovarian cancer stem cell growth under the action of fructose. However, the mechanism by which CMA participates in the promotion effect has not been elucidated [55].

CMA exerts a procancer effect by preventing the degradation of tumor-associated proteins. The pro-survival protein MCL1 is one of the critical components of lung cancer cell survival, and studies have demonstrated the presence of a CMA-mediated MCL1 protein stabilization system in lung cancer cells, which may contribute to cancer progression [56]. The latest study showed that impaired degradation of two proteins, the transcriptional co-activator *YAP1* and *IL6ST* (interleukin six cytokine family signal transducer), by CMA promotes tumor growth, and it demonstrated that *YAP1* and *IL6ST* are novel substrates for CMA [57]. CMA regulation of the proteome in cancer continues to provide us with new therapeutic strategies. It is worth considering as a future direction the possibility of controlling CMA activity through proteins as a reference strategy to inhibit tumorigenesis.

Although high levels of CMA activity have a procancer function in cancer cells, they can also promote cancer cell development by deactivating tumor suppressors, pro-apoptotic factors, and anti-proliferative factors through CMA [42]. An increasing number of studies have demonstrated that various tumor suppressors can be substrate proteins in the CMA pathway, including mainly p73, N-CoR, PED, and RND3, among others, and that the procancer effects of CMA are achieved by degrading these tumor suppressors [58]. Ali et al. found that nuclear receptor co-repressor (N-CoR) is relevant to the chaperone molecule HSPA8 and that the CMA pathway promotes the survival and growth of NSCLC cells by degrading misfolded N-CoR [59]. Kon et al. found that blocking the expression of LAMP-2A in *mice* was able to reduce the metastatic ability of lung cancer cells [6]. Recent studies have shown that high expression of LAMP-2A in glioblastoma multiforme (GBM) inhibits apoptosis by downregulating N-CoR, thereby promoting tumor growth [60]. Similarly, LAMP-2A is also overexpressed in breast tumors and consequently promotes the survival of cancer cells [53]. Furthermore, Ding et al. found that LAMP-2A expression was significantly increased in HCC tissues compared to normal liver tissues, and blocking LAMP-2A could inhibit HCC cell viability, suggesting that the presence of LAMP-2A is a necessary condition for tumor growth [23]. Therefore, downregulating the expression mechanism of LAMP-2A in tumors has positive significance for tumor treatment.

*PED* is an anti-apoptotic molecule that occurs in two forms in cells: phosphorylated *PED* (as a tumor promoter) and unphosphorylated *PED* (as a tumor suppressor) [61]. Quintavalle et al. demonstrated that *PED* is not only a bona fide CMA substrate but also that it interacts with HSPA8. Unphosphorylated *PED* in lung cancer is degraded by targeting lysosomes through the CMA pathway, which may promote cancer progression [62]. In addition, Nguyen et al. discovered a novel mechanism: nerve growth factor receptor (NGFR) can promote the degradation of *p73* through the CMA pathway. *P73* is one of the tumor suppressors of the *p53* family [63]. Therefore, by weakening the inhibitory effect of *p73* on cancer cell growth through NGFR, we found that the new mechanism of CMA positively affects cancer cell survival. CMA degrades the antiproliferative protein RND3 (Rho family GTPase 3) to maintain the rapid proliferation of gastric cancer cells [64]. The CMA pathway can also exert procancer effects by indirectly degrading tumor suppressors. For example, sorting nexin 10 (SNX10) deficiency promotes colorectal cancer cell (CRC) proliferation through activation of the CMA pathway, which results in the degradation of the tumor suppressor *p21* [65]. SNX10 deficiency promotes tumor development in colorectal cancer, so altering the downregulated activity of SNX10 may be beneficial in treating colorectal cancer. In conclusion, the procancer effects exhibited by CMA have undoubtedly provided new targets for human cancer therapy.

Compared with normal cells, tumor cells still produce ATP through aerobic glycolysis to meet their energy consumption needs, even when the oxygen supply is sufficient. This phenomenon is known as the Warburg effect [66]. Pyruvate kinase isoform M2 (PKM2) is an essential enzyme in the glycolysis process. The acetylation of PKM2 increases its interaction with HSPA8 and promotes lysosome-dependent degradation of PKM2 through CMA, which promotes tumor growth [67]. In renal cell carcinoma, CMA can inhibit the apoptosis of cancer cells through PKM2, but the specific mechanism has not been shown [68]. It is worth mentioning that the development of Warburg effect inhibitors has opened up ideas for improving cancer treatment. Another new finding reveals that under glucose starvation, a novel regulator, PIM2 (proviral insertion in murine lymphomas 2), regulates the phosphorylation of HK2 through a CMA-mediated protein degradation pathway, stabilizing the protein to promote glycolysis. This promotes proliferation, migration, and tumor growth in breast cancer cells [69]. Similarly, studies in breast cancer have shown that elevated CMA activity can promote cancer cell metastasis by downregulating autophagy-related gene 5 (*ATG5*)-dependent macroautophagy pathway activity [70,71]. Enhanced expression of stromal cell-derived factor 1 (SDF1) and C-X-C motif chemokine receptor 4 (CXCR4) in the tumor microenvironment can promote tumor proliferation and migration. It has been shown that CMA enhances the expression of SDF1 and CXCR4 by degrading peroxisome proliferator-activated receptor γ (PPAR γ) and promotes the development of thyroid cancer [72]. This also suggests that understanding the link between cellular chemotaxis and CMA may help to create antitumor therapies.

The current views on CMA in existing cancers are mostly biased toward its cancer-promoting role. However, for cancer treatment, there are still breakthroughs and challenges in studying the CMA pathway and finding ways to reduce the growth and metastasis of cancer cells by blocking the cancer-promoting mechanism (Table 1).

## 4. Prevention and Treatment of CMA in Cancer

Despite the complex links between CMA and cancer biology, we can still find breakthroughs in cancer treatment when the roles and limitations of the various influencing factors are individually analyzed. LAMP-2A is a breakthrough in current therapeutic drug development and is critical to the CMA pathway [77]. This suggests that restoring CMA activity to normal cellular levels in the aging liver could improve cellular maintenance of liver function.

### 4.1. Prevention

In addition to treating cancers that have already occurred, several studies have suggested that preventive measures can be taken before they occur. Zhang et al. has found that genetic prophylaxis of age-dependent reduction of CMA in *mouse* liver attenuates hepatic protein water toxicity and reduces the accumulation of cellular damage. The positive effect of this intervention on liver homeostasis may prevent the malignant transition of this organ [78]. Given the anticancer activity of CMA in healthy cells, stabilizing CMA activity may be a cancer prevention therapy.

Because the odds of tumorigenesis are associated with a decrease in CMA activity with age, it is not difficult to explain the greater incidence of cancer in older age groups. For example, in an earlier study of CMA-deficient *mouse* liver, Gomes et al. found disrupted protein homeostasis and reduced stress capacity, increasing the probability of liver tumorigenesis in CMA-deficient aged *mice* [43]. When referring to the relationship between CMA and aging, it is worth noting that CMA is related to the nutritional status of aging organisms [77]. It is hypothesized that nutritional interventions to address nutrient deficiencies in aging organisms could maintain CMA activity to prevent malignant transformation. Surprisingly, the increase in CMA activity in senescent cells was first reported in 2022 by Rovira et al., who summarized the phenomenon of upregulation of the CMA pathway in senescent cells by examining five cell types; they also found an increase in macroautophagy in senescent cells [29]. This paradox forces us to consider whether we should increase CMA activity or suppress CMA activity to prevent cancer development in older populations when treating cancer. We suggest that the changes in protein degradation due to alterations in the macroautophagy and CMA pathways in senescent cells remain to be investigated. Clarifying the mechanisms connecting CMA and protein degradation is the first step towards preventing cancer through the CMA pathway.

### 4.2. Treatment

Indeed, in various cancer cell lines, downregulation of LAMP-2A resulted in a significant reduction in proliferation, which not only sensitized the cells to stressors but also reduced the ability of xenografts to form cancer cells. Thus, the antitumor effect of CMA downregulation could be used as a cancer treatment strategy [23,53]. In another study, multiple metabolic proteins were targeted for degradation by the CMA pathway, which induces cancer cell death through metabolic catastrophe [79]. In summary, selective activation of CMA could be a therapeutic strategy to eliminate mutant proteins associated with human cancers.

Most recent studies suggest inducing CMA during cancer treatment may be a key mechanism for drug resistance in different types of cancer. For example, Cao et al. suggested that the downregulation of CMA activity in patients with esophageal squamous cell carcinoma (ESCC) inhibited ESCC cell proliferation and colony formation, with increased sensitivity to therapeutic agents [7]. Resistance to sorafenib in hepatocellular carcinoma poses a significant challenge to drug treatment in patients. However, research has indicated that under hypoxic conditions, hsp90α binds to necrosome complexes and undergoes degradation via the CMA pathway. This alteration in the distribution of necroptosis-related proteins may enhance cancer resistance to sorafenib [80]. CMA receptors also play an active role in regulating tumor growth and chemoresistance in non-small-cell lung cancer [8]. In addition, exosomes derived from HBV-associated hepatocellular carcinoma cells promote chemoresistance by regulating the CMA pathway [81]. By investigating the mechanisms of drug resistance in multiple cancer therapies, we believe that the role of the CMA pathway in drug resistance can be exploited as a therapeutic strategy for cancer. In conclusion, it is critical to find small molecules in the CMA pathway involved in the mechanism of resistance to cancer therapy, which has practical significance for the research direction of eliminating drug resistance in cancer cells.

Moreover, the study revealed that the involvement of CMA drives the balance between stem cell proliferation and cell differentiation, which illustrates the close relationship between cancer and the activities performed by these two types of cells [82]. It can be posited that the investigation of CMA in stem cells may offer a novel approach to cancer therapy. The emergence of CMA as a targeting strategy for cancer therapy is evident, and the inhibition of tumor growth, metastasis, or resistance through the blockading of CMA may serve as a potential treatment for established tumors. Nevertheless, the restricted comprehension and awareness of the physiological regulatory pathway of CMA present numerous impediments to the utilization of targeting compounds and selective chemo-modulators of this pathway. Furthermore, different cancer types and various stages of development are worth considering as influential factors (Figure 2).

## 5. Conclusions and Outlook

The activity of CMA is directly related to the function of LAMP-2A, which is on the lysosome membrane and is an essential part of the CMA pathway. In addition, CMA activity can also be regulated by mediating the levels of molecular chaperones and substrate proteins. However, it is yet unknown exactly how molecular chaperones other than HSPA8 contribute to the control of CMA. As CMA is predominantly upregulated in cancer, most of the research on CMA and cancer has focused on its procancer mechanisms. However, it is imperative to direct greater attention toward CMA that assumes an oncogenic role in cancer. As such, promoting the oncogenic mechanism in cancer therapy and considering the dual role of CMA are essential.

Although the regulation of CMA activity in various tumors has been preliminarily explored, there is still a lack of effective potential drug targets in tumor therapy. It becomes difficult to develop effective CMA modulators due to the crosstalk between CMA and other forms of autophagy and its complex physiological functions. We need to develop selective inhibitors of CMA, not just ones that disrupt all autophagy pathways by hydrolyzing proteins. It is posited that an investigation into the function of CMA in different stages of tumor progression may yield targeted therapeutic interventions. Additionally, we believe investigating the transition of CMA from an oncogenic effect in non-transformed cells to a tumor-promoting function in cancer cells is a promising avenue in cancer treatment.

## Figures and Tables

**Figure 1 biomedicines-11-02050-f001:**
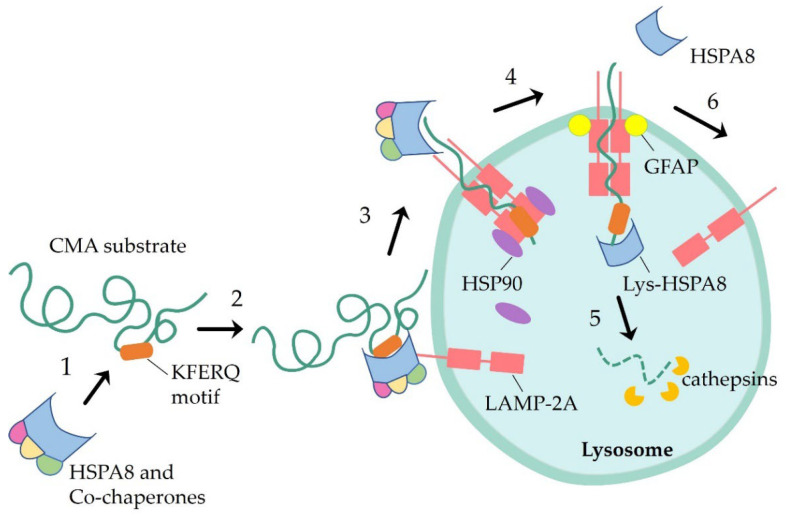
CMA occurs: (1) The molecular chaperone HSPA8 recognizes the corresponding substrate protein and transports it to the lysosome. (2) The receptor LAMP-2A located on the surface of the lysosomal membrane binds to the substrate protein, and HSPA8 is involved in the unwrapping of the substrate protein. (3) Hsp90 binds the receptor LAMP-2A to promote and stabilize its multimerization, thus forming a high-molecular-weight complex. (4) The substrate complex is assisted by molecular chaperones (lysosomal HSPA8) on the luminal side of the molecular membrane with translocation across the lysosomal membrane; GFAP interacts with LAMP-2A to stabilize the complex required for translocation. (5) The substrate protein is hydrolyzed by protease. (6) LAMP-2A is reformed as a monomer ready to enter the next binding. (GFAP, glial fibrillary acidic protein; lys-HSPA8, lysosomal HSPA8).

**Figure 2 biomedicines-11-02050-f002:**
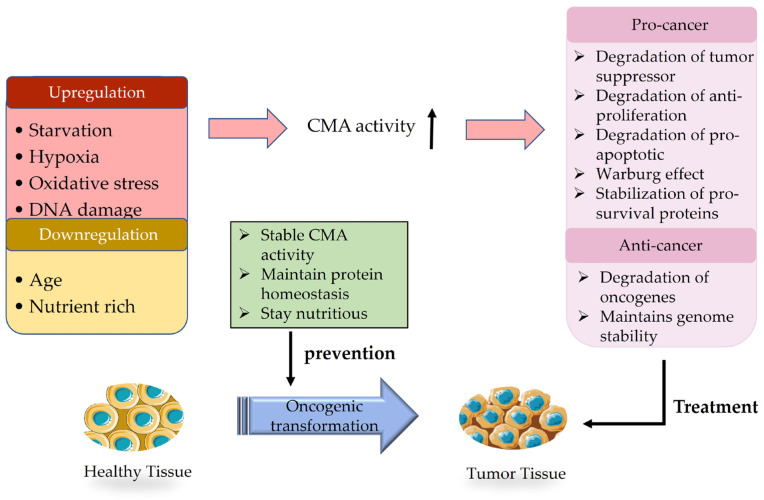
The impact of the CMA pathway on cancer and its prevention and treatment. CMA activity is altered by several factors (upregulation: starvation, hypoxia, oxidative stress, DNA damage; downregulation: age, nutrient richness), and it generally exhibits upregulated activity in tumor tissues. On the one hand, highly active CMA can promote tumor growth by degrading tumor suppressors and other factors. On the other hand, highly active CMA can inhibit tumor growth by degrading oncogenic factors, etc. Taking appropriate measures (such as maintaining protein homeostasis and so on) can prevent healthy cells from malignantly transforming into tumor cells. The dual effects of CMA on cancer were used to treat cancer tissue.

**Table 1 biomedicines-11-02050-t001:** The role of CMA in different cancers.

Type of Cancer	CMA Action Factor and Its Activity	CMA Activity	Cellular Physiological Processes	Subjects	Cancer Development	Reference
Pancreatic carcinoma	IGF-IR, Down	Up	Cell signaling	The human PC Cell lines	Inhibition	[73]
Ovarian carcinoma	HK2, Down	Up	Glucose metabolism	Cell lines	Inhibition	[48]
Breast carcinoma	HSD17B4, Down	Up	Protein modification	Breast cancer tissues	Inhibition	[52]
	*ATG5*, Down	Up	Gene transcription	Breast cancer Cell lines	Promotion	[70]
Hepatocellular carcinoma	HMGB1, Down	Up	Gene expression	Cell lines	Promotion	[74]
	*TP53*, Down	Up	Gene expression	Cell lines	Inhibition	[49]
Colorectal carcinoma	*P21*, Down	Up	Gene expression	CRC tissues	Promotion	[65]
Gastric carcinoma	RND3, Down	Up	Cell growth	GC tissues	Promotion	[64]
Papillary thyroid carcinoma	PPAR γ, Down	Up	Cell signaling	PTC tissues	Promotion	[72]
Glioblastoma	N-CoR, Down	Up	Protein modification	Glioma cell line U87-MG and tissues	Promotion	[60]
Lung carcinoma	N-CoR, Down	Up	Protein folding	Lung cancer cell lines	Promotion	[59]
	ErK3, Down	Up	Protein modification	Cell lines	Inhibition	[75]
	MCL1, Up	Down	Protein modification	NSCLC cell lines	Promotion	[56]
Renal carcinoma	PKM2, Up	Down	Glucose metabolism	Cell lines	Promotion	[68]
Prostate carcinoma	TPD52, Up	Down	Protein modification	C57BL/6 mouse	Promotion	[76]

## Data Availability

Not applicable.

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
