# Peer review of "The Complex Role of Chaperone-Mediated Autophagy in Cancer Diseases"

_biomedicines, 2023, doi:10.3390/biomedicines11072050_

Round 1
Reviewer 1 Report
In the recent years, several extensive reviews and many articles have appeared on autophagy in relation with immunity and pathophysiology. Analysing this large and important literature represents a lot to read for beginners in the field or for those researchers who are interested by a concise overview of the field. There are also some debates in the current publications about the place of CMA in the complex network of autophagy processes that deserve a careful look and analyses in depth. With regard to this aspect, the paper submitted by Li et al., which is focused on CMA in cancer diseases, could have been pertinent. In the present article, however, the elements of discussion are not extensive and some major articles published recently are not cited. However, we note that at least 23/69 publications of one single (recognized leader) team are cited. We can also regret that discrepant results reported in the literature are just superficially described but not further discussed. Below are specific concerns regarding this manuscript:
1. The paper starts with a classical format of description found in many past reviews (including the Fig. 1 that can be seen in many reviews -here a reproduction- and is incomplete in 2023; and obviously figure 2 who is also a reproduction). The sections 1 and 2 contain very common information that could be summarized with references to more detailed articles, including those centred on lysosomal dysfunction in cancer diseases. Many sentences are just repeated from other articles. It is regrettable that the authors did not choose to present these themes from another, original and more personal angle.
2. Abstract: the meaning of the sentence “This review aims to provide some feasible references” is not clear. Is the purpose of this Review just to list references?
3. Instead of the common Figures 1 and 2, at least one original figure reporting the effects of CMA in cancer diseases should be inserted to highlight specific key points on which the authors wish to focus this review. This would enormously enrich the article and serves the Authors.
4. The nomenclature “HSPA8” should be used instead of the old name “HSC70”.
5. Information based on cancer cells should not appear in the section 2.2.
6. “Cancer” is often use as a whole while we all know that autophagy defects can be differently regulated (activated, down-regulated) according to the type of cancer, in different organs and tissues. For this review to be useful and informative to the reader, it would be important each time to indicate the type of cancer or tumour cells, if the study is done in human or animal context, in vivo or in vitro, with cell lines (and which one). The table 1 is oversimplified.
7. The complex links between autophagy (and its deregulation) and cell chemotaxis, especially in cancer, is not analysed in depth. The section 3.3 is largely incomplete.
8. The link between PARP, an enzyme involved in DNA repair and cancer, and CMA is not described.
9. Section 4: a scheme of possible ways (targets and tools) of intervention should be inserted to illustrate what is presented in a narrative style. This section, probably one of the most important for the reader, is superficial. Sentences are common, there is no original path of worthwhile investigation.
10. In the conclusion, the authors might better highlight what they bring in terms of intellectual knowledge compared to previous reports and avoid repeating certain common ideas.
11. Typos and mistakes should be corrected: for example, line 86 (LyC), lines 100 (Hsc40 and Hsc90), others.
12. All over the article several abbreviations are not described.
Author Response
We feel great thanks for your professional review work on our manuscript. In the following pages are our point-by-point responses to each of your comments. Thank you again for your time, effort, and very helpful comments, which have helped us to improve and perfect our manuscript.
- Response: Thank you very much for your comments. We have replaced our creation diagram in the revised manuscript. We feel very guilty about our extensive and insightful overview, and we have updated and referred to additional articles to supplement the organization of our manuscript.
- Response: Thank you very much for your comments and suggestions. We have corrected the inappropriate description in the abstract. We fully agree with your opinion and have added corresponding discussions in the revised manuscript to enrich the main body of the manuscript.
- Response: Thanks for your advice. We have inserted Figure 2 in the revised manuscript to highlight the impact of CMA on cancer disease.
- Response: Thanks for your advice. We have replaced all of 'HSC70' with 'HSPA8' following your advice.
- Response: Thank you very much for your suggestion. We have removed the information about cancer cells in section 2.2 in the revised manuscript.
- Response: We are very appreciative of your comments and suggestions. We have added the corresponding information in Table 1, including the activity of CMA and the subject of the study.
- Response: Thank you very much for your suggestions. We apologize for our lack of in-depth analysis. As our review focuses on CMA, we have added the relationship between CMA and cellular chemotaxis in cancer in lines 388-393 of the revised manuscript. Considering the structure of the manuscript, we have integrated the content of 3.3 into sections 3.1 and 3.2, respectively.
- Response: Thanks for your advice. We have added information about the link between CMA and PARP (lines 157-159).
- Response: Thank you very much for your advice. We apologize for superficially writing this section. We have made adjustments in the revised manuscript, and relevant instructions on cancer prevention and treatment strategies are shown in Figure 2.
- Response: Thank you very much for pointing out our shortcomings. We have revised and rewritten the conclusions in the revised manuscript.
- Response: Thanks for your comment. We are sorry for the mistakes. We have corrected the spelling mistakes (lines 80,96).
- Response: Thanks for your comment. We have described the corresponding abbreviations.
We appreciate your warm work earnestly and hope that our corrections will be recognized. Once again, thank you very much for your comments and suggestions.
Reviewer 2 Report
In this manuscript, the authors explore the role of chaperone-mediated autophagy (CMA) in cancer. It is a timely, interesting and well written manuscript that explores several aspects of carcinogenesis and its interplay with CMA. However, some points need to be clarified before this manuscript can be accepted for publication.
My specific comments:
In the abstract, it says to terminate cellular function. Please replace with: to terminate their cellular function, or any similar sentence which does not suggest induction of cell death.
Line 29, the authors state that: As early as 26 the 1970s, it has been studied that there are two pathways for protein degradation: the ATP-dependent ubiquitination pathway and the ATP-independent lysosomal degradation pathway. This is not entirely true, as the review that the authors cite (2) mentions: “The energy requirement for protein degradation was described as indirect, and necessary, for example, for protein transport across the lysosomal membrane and/or for the activity of the H+ pump and the maintenance of the low acidic intralysosomal pH that is necessary for optimal activity of the proteases.” Please clarify since lysosomal degradation is not ATP-independent.
Line 38, it says “Macroautophagy is the most common form found in 37 mammals and is particularly active in cases of nutritional starvation”. Although macroautophagy is increased upon nutritional starvation, other stimuli induce macroautophagy and it also occurs at basal levels in the cells with the purpose of maintaining cytoplasmic fitness. Please clarify.
Line 86, it says two positively charged aminoacids (Lyc, Arg), please correct.
Throughout the text and figures, the authors use Hsc70/HSC70 or HSP90/Hsp90, please homogenize. Also, in the figures, Hsp90 and HSC70 are labeled as LysHSP90 and LysHSC70, what does Lys stand for? It is not clear in the text or in the figure legends.
Section 3.3. it says: “Since CMA blockade decreases anal migration and resistance and significantly reduces the metastatic potential of lung cancer cells”, what does anal migration mean? The manuscript only evaluates lung metastasis. Please clarify.
Line 335, what are ATG5-dependent macrophages? Please clarify.
Section 3.4 is entitled: Mechanism of CMA in cancer, however, the previous sections are also related to CMA and cancer. Please change the section title according to the subthemes.
Line 353, it says: “Although the specific processes of CMA regulation by cells in the tumor 353 microenvironment are not well expressed, the proposal of microtubule protrusions and 354 tumor-generated exosomes makes this possible”. This sentence is not clear, please clarify.
Section 3.4.2. Regulation of CMA and proteome includes examples that could be incorporated in sections 3.1 and 3.2 where the authors mentions pro- and anti-tumorigenic proteins degraded by CMA. Is this section necessary or can it be eliminated and incorporated in sections 3.1 and 3.2?
Line 362, please define GSC.
Section 4.2. the authors support the tumor suppressive role of CMA in the fact that CMA activity declines with age. However, they do not address other types of autophagy? Is macroautophagy also decreased upon aging? What would be the contribution of each type of autophagy to the increased incidence of cancer in aging patients?
Minor revisions to English Language are needed.
Author Response
We feel great thanks for your professional review work on our manuscript. In the following pages are our point-by-point responses to each of your comments. Thank you again for your time, effort, and very helpful comments, which have helped us to improve and perfect our manuscript.
1) Response: Thanks for your advice. We didn't think it through enough and have corrected it in the revised manuscript as you suggested.
2) Response: Thanks for your comment. First of all, we are very sorry for our loose statement. Secondly, we strongly agree with your advice and have made corrections in line 29.
3) Response: Thanks for your advice. In response to your reminder, we have corrected the statement about macroautophagy.
4) Response: We are sorry for the mistakes. We have corrected the spelling in line 86.
5) Response: Thank you very much for your suggestion. We have harmonized the expressions. 'Lys' stands for the abbreviation of the lysosome, which we have indicated in the legend of the revised manuscript.
6) Response: We apologize for the misrepresentation and thank you for your comment. We wanted to reorganize part 3.3, so we made some adjustments. We increased metastasis to other types of tumors in sections 3.1 and 3.2, respectively.
7) Response: Thanks for your advice. We are sorry for the unsuitable description. We rephrased it in the revised manuscript as 'autophagy-related gene 5(ATG5)-dependent macroautophagy pathway activity'.
8) Response: We are very appreciative of your comments and suggestions. As you mentioned, this section overlaps with the previous section, so we have reorganized the structure of the article and adjusted this section to section 2.3.
9) Response: Thanks for your advice. We are sorry for the unsuitable description. We have deleted the statement in the revised manuscript. Considering the impact of the tumor microenvironment on cancer, we redescribe its related content in Part 3.
10) Response: We are very appreciative of your comments and suggestions. It is a pity that we did not consider the inner connection of the article. We have included this section in 3.1 and 3.2, respectively.
11) Response: Thanks for your advice. We have added the definition of GSC in line 259.
12) Response: Thank you very much for your review and suggestions. As our review focuses on CMA, we regret that we overlooked other types of autophagy. Thank you very much for your reminder, and we have reviewed the literature carefully to add a link between macroautophagy and aging in lines 205-210.
We appreciate your warm work earnestly and hope that our corrections will be recognized. Once again, thank you very much for your comments and suggestions.